# Associations between frontal lobe structure, parent-reported obstructive sleep disordered breathing and childhood behavior in the ABCD dataset

Amal Isaiah [1,2✉], Thomas Ernst[3,4,5], Christine C. Cloak[3], Duncan B. Clark[6] & Linda Chang [3,4,5,7]

Parents frequently report behavioral problems among children who snore. Our understanding of the relationship between symptoms of obstructive sleep disordered breathing (oSDB) and childhood behavioral problems associated with brain structural alterations is limited. Here, we examine the associations between oSDB symptoms, behavioral measures such as inattention, and brain morphometry in the Adolescent Brain Cognitive Development (ABCD) study comprising 10,140 preadolescents. We observe that parent-reported symptoms of oSDB are associated with composite and domain-specific problem behaviors measured by parent responses to the Child Behavior Checklist. Alterations of brain structure demonstrating the strongest negative associations with oSDB symptoms are within the frontal lobe. The relationships between oSDB symptoms and behavioral measures are mediated by significantly smaller volumes of multiple frontal lobe regions. These results provide population-level evidence for an association between regional structural alterations in cortical gray matter and problem behaviors reported in children with oSDB.

[1] Department of Otorhinolaryngology—Head and Neck Surgery, University of Maryland School of Medicine, Baltimore, MD, USA. [2] Department of Pediatrics, University of Maryland School of Medicine, Baltimore, MD, USA. [3] Department of Diagnostic Radiology and Nuclear Medicine, University of Maryland School of Medicine, Baltimore, MD, USA. [4] Department of Medicine, John A. Burns School of Medicine, University of Hawaii at Manoa, Honolulu, HI, USA. [5] Department of Neurology, Johns Hopkins University School of Medicine, Baltimore, MD, USA. [6] Department of Psychiatry, University of Pittsburgh School of Medicine, Pittsburgh, PA, USA. [7] Department of Neurology, University of Maryland School of Medicine, Baltimore, MD, USA. ✉email: aisaiah@som.umaryland.edu

Obstructive sleep disordered breathing (oSDB) in children is characterized by resistance to airflow in the upper airway, resulting in snoring and difficulty breathing with or without respiratory pauses during sleep[1]. Untreated oSDB is commonly associated with neurobehavioral deficits[2] justifying its diagnosis and management[3]. With an overall prevalence of 5-10%[4], it is also the principal indication for surgical removal of tonsils and adenoids—a procedure performed in hundreds of thousands of children worldwide[5].

The importance of parent-reported symptoms to screen for oSDB is highlighted by large, national, population-based studies demonstrating a strong bidirectional relationship between oSDB symptoms and problem behaviors such as inattention, hyperactivity and aggression. For example, in the Avon Longitudinal Study of Parents and Children comprising 8,000 children from Southwest England, the key symptoms of oSDB, especially snoring, predicted problem behaviors[6]. Similar relationships were identified between habitual snoring and hyperactivity in a survey of 6000 children from Hong Kong using parent-reported measures[7]. These results have been further replicated by population-based assessments from Portugal[8], Turkey[9], and Argentina[10]. Furthermore, given the widely described relationship between oSDB and behavioral measures, parent-reported symptoms of upper airway obstruction are also central to screening for conditions such as obstructive sleep apnea[3], as well as the surgical decision making in hundreds of thousands of children undergoing tonsillectomy and adenoidectomy each year[11].

Although the terminology has varied over the years, habitual snoring, defined as snoring occurring three or more nights a week, is considered a key component of oSDB[7,12], and a predictor of the associated problem behaviors in children[9,12,13]. The consistent relationship between oSDB and problem behaviors involving deficits in cognitive control, such as inattention and hyperactivity[14,15] may impact children's learning and interaction in the classroom[16]. However, ascertaining this poorly understood link between oSDB and problem behaviors in previous population-based studies requires the interrogation of brain-related changes common to both[17]. Evidence of brain structural changes in children with oSDB have been limited to a few small studies of associations between obstructive sleep apnea and alterations in brain tissue or neuronal integrity[18–20]. The changes within the cortical gray matter defining the relationship between oSDB and behavioral measures in large population-based cohorts remain unverified. Validating this concurrent negative relationship also requires robust statistical control for confounding from demographic and socioeconomic factors that often impact the generalizability of small studies[21,22]. Cumulatively, the results from previous studies highlight two major facets of the gaps in our knowledge derived from large investigations of behavior in children with oSDB. First, no population-based neuroimaging study has investigated whether snoring in children is associated with underlying neuroanatomic alterations following statistical control for common confounders such as socioeconomic status. Second, no study, large or small, has investigated whether the association between oSDB and behavioral measures in children could be attributable to structural alterations in the brain. Addressing these gaps could streamline the evaluation and management of children with symptoms of oSDB by better delineating the pathophysiology of oSDB-related neurobehavioral morbidity, such as problem behaviors and poor school performance, identifying children at risk for the morbidity based on symptoms and enabling clinicians to monitor clinically significant post-treatment recurrence of oSDB in children.

The Adolescent Brain Cognitive Development (ABCD, https://abcdstudy.org) Study is a long-term study of brain development and child health in the United States and comprises a diverse sample of typically developing children. Given the importance of oSDB with a prevalence approaching 10%[4], we determined whether the observations from previous studies concerning the regional changes in the brain related to oSDB are replicable in the ABCD dataset. We hypothesized that the associations between parent-reported symptoms of oSDB and behavioral measures in children are mediated by alterations in brain morphometric characteristics.

Here, we show that the symptoms of oSDB, such as snoring, are associated with composite and domain-specific problem behaviors in children assessed using child behavior checklist (CBCL). Multiple frontal lobe regions are smaller in volume among children with greater symptom burden of oSDB. The regional frontal lobe volumes additionally mediate the relationship between parent-reported oSDB and problem behaviors measured in children using CBCL. These findings support a potential neural substrate at a population level for the previously observed behavioral morbidity in children with symptoms of upper airway obstruction.

## Results

**Baseline characteristics.** Out of 11,875 children, 10,140 were included in the analysis following removal of incomplete data and imaging files that failed quality control. Table 1 summarizes the baseline variables: 5300 children (52.3%) were boys, 5547 (54.7%) were white, and 1567 (15.5%) were obese. Habitual snoring, defined as snoring at least three nights a week, was reported in 661 children (6.5%). In addition, snoring was the most predominant symptom of oSDB reported by the parents (Fig. 1a–c). A total of 1726 (17.0%) children had a diagnosis of asthma. The distributions of household income and highest caregiver education status are also shown in Table 1. The mean ± SD t-scores for the CBCL categories of total, externalizing and internalizing problems were 45.8 ± 11.2, 45.6 ± 10.2, and 48.5 ± 10.6, respectively. The eight category scores are summarized in Supplementary Table 1. The proportion of children with abnormal CBCL scores, defined using established thresholds[23], are shown in Supplementary Table 1. For the internalizing problems scale, 1694 children, or 16.7%, were classified as abnormal based on the threshold t-score of 60 or higher.

The associations between each of the demographic factors and the oSDB factor score, representing the overall burden of oSDB from the three symptoms of snoring, breathing problems and respiratory pauses, is summarized in Supplementary Table 2. The greatest effect sizes were observed for body mass index percentile score ($F_{(1, 23.57)} = 119.60$, $\Delta R^2_{\text{adjusted}} = 1.11$, $P < 10^{-16}$), a history of asthma ($F_{(1, 23.57)} = 108.41$, $\Delta R^2_{\text{adjusted}} = 0.98$, $P < 10^{-16}$) and race/ethnicity ($F_{(4, 23.57)} = 21.82$, $\Delta R^2_{\text{adjusted}} = 0.70$, $P < 10^{-16}$). The effect sizes for age ($F_{(1, 23.57)} = 6.40$, $\Delta R^2_{\text{adjusted}} = 0.05$, $P = 0.011$), sex ($F_{(1, 23.57)} = 1.46$, $\Delta R^2_{\text{adjusted}} = 0.00$, $P = 0.23$), household income ($F_{(3.46, 23.57)} = 8.18$, $\Delta R^2_{\text{adjusted}} = 0.23$, $P = 9.3 \times 10^{-6}$) and highest parental educational status ($F_{(1, 23.57)} = 6.45$, $\Delta R^2_{\text{adjusted}} = 0.05$, $P = 0.011$) were much smaller. These relationships were also observed for individual symptom scores (Supplementary Table 2).

**Problem behaviors.** The oSDB factor score predicted the total problems CBCL scale following adjustment for covariates ($F_{(1, 35.96)} = 334.14$, $\Delta R^2_{\text{adjusted}} = 2.96$, $P < 10^{-16}$; Fig. 2a). Similar relationships were observed for both externalizing ($F_{(1, 34.61)} = 185.27$, $\Delta R^2_{\text{adjusted}} = 1.68$, $P < 10^{-16}$; Fig. 2b) and internalizing problems ($F_{(1,34.37)} = 243.18$, $\Delta R^2_{\text{adjusted}} = 2.20$, $P < 10^{-16}$; Fig. 2c). These relationships were independent of sex (Fig. 1d–f) and are also shown for individual CBCL scales (Supplementary

**Table 1 Baseline characteristics of the Adolescent Brain Cognitive Development Study ($n = 10{,}140$).**

| Variable | Value |
|---|---|
| Age (months) | 118.9 ± 7.5 |
| *Sex* | |
| Female | 4840 (47.7%) |
| Male | 5300 (52.3%) |
| *Race/ethnicity* | |
| White | 5547 (54.7%) |
| Black | 1972 (19.5%) |
| Hispanic | 1376 (13.6%) |
| Asian | 1051 (10.4%) |
| Other | 194 (1.9%) |
| BMI percentile | 59.2 ± 31.2 |
| Obesity | 1567 (15.5%) |
| History of asthma | 1726 (17.0%) |
| *Total household income before taxes (US Dollars)* | |
| <5000 | 372 (3.7%) |
| 5000–12,000 | 387 (3.8%) |
| 12,000–16,000 | 251 (2.5%) |
| 16,000–25,000 | 490 (4.8%) |
| 25,000–35,000 | 616 (6.1%) |
| 35,000–50,000 | 858 (8.5%) |
| 50,000–75,000 | 1408 (13.9%) |
| 75,000–100,000 | 1484 (14.6%) |
| 100,000–200,000 | 3112 (30.7%) |
| >200,000 | 1162 (11.5%) |
| *Highest education status of the caregiver* | |
| <High school graduate | 540 (5.3%) |
| High school graduate | 765 (7.5%) |
| High school/General Educational | 982 (9.7%) |
| Development | |
| Some college/associate | 2964 (29.2%) |
| Bachelor's degree | 2956 (29.2%) |
| Master's degree | 2046 (20.2%) |
| Professional/doctoral degree | 652 (6.5%) |
| *Snoring frequency* | |
| Never | 6040 (59.6%) |
| Non-habitual (snoring 1–2 nights a week) | 3439 (33.9%) |
| Habitual (>2 nights/week) | 661 (6.5%) |

Categorical variables are described by number (%) and continuous variables by mean ± standard deviation. Sex refers to the biological sex at birth. Race/ethnicity was self-selected by the primary caregiver. Body mass index (BMI) percentile score was derived by first calculating the body mass index after dividing the weight in kilograms by the square of height in meters and derived from Centers for Disease Control growth charts. Obesity was defined as BMI ≥ 95th percentile. The frequency of parent-reported snoring was considered non-habitual if it occured 1–2 nights/week and habitual if more than 2 nights/week.

Fig. 1). Furthermore, for individual oSDB symptoms, the strongest association was identified between the frequency of snoring and higher CBCL scores (Fig. 1g). Higher household income partially mitigated the association between oSDB symptoms and CBCL scores (Fig. 2). The magnitudes of these interactions were similar across individual CBCL scales and for every oSDB symptom (Fig. 2k).

**Brain morphometry.** Assessment of the relationships between oSDB factor score and brain morphometric measures demonstrated a small effect size for total cortical volume ($F_{(1, 37.98)} = 12.88$, $\Delta R^2_{adjusted} = 0.08$, $P = 0.0003$) and smaller effect sizes for average cortical thickness ($F_{(1, 39.46)} = 5.77$, $\Delta R^2_{adjusted} = 0.03$, $P = 0.016$), and cortical surface area ($F_{(1, 43.47)} = 7.34$, $\Delta R^2_{adjusted} = 0.04$, $P = 0.006$). Covariates included age, sex, race/ethnicity, and household pre-tax income as fixed effects and scanner ID as random effect. The greatest negative association among the three symptoms of oSDB with morphometric measures was observed between the frequency of snoring and cortical thickness (Fig. 3c).

The composite oSDB factor score demonstrated a stronger relationship with average cortical thickness than any of the individual symptoms of oSDB (Fig. 3d). The top ten cortical regions arranged by their effect size for the association of the factor, as well as individual oSDB symptom scores with each of the three morphometric variables are shown in Fig. 3e. Most regions demonstrating these negative associations were located within the frontal lobe. For example, the oSDB factor score predicted a thinner left medial orbital sulcus ($F_{(1, 38.30)} = 4.76$, $\Delta R^2_{adjusted} = 0.27$, $P = 1.76 \times 10^{-5}$) and a smaller left precentral gyrus ($F_{(1, 33.76)} = 14.44$, $\Delta R^2_{adjusted} = 0.10$, $P = 1.47 \times 10^{-4}$). Among the individual symptoms, the frequency of snoring similarly predicted a thinner left medial orbital sulcus ($F_{(1, 38.28)} = 4.35$, $\Delta R^2_{adjusted} = 0.24$, $P = 6.47 \times 10^{-5}$) and a smaller left precentral gyrus ($F_{(1, 33.77)} = 16.03$, $\Delta R^2_{adjusted} = 0.10$, $P = 6.23 \times 10^{-5}$). The dose-dependent associations between the oSDB factor scores and the frequency of snoring as predictors, and the mean thickness of the left medial orbital sulcus are further illustrated in Fig. 3f, g. To further explore a clinical threshold for these morphometric alterations, we grouped the frequency of snoring as non-habitual snoring (snoring reported less than three nights a week) and habitual (at least three nights a week). These pairwise comparisons, as shown in Fig. 3h, i identified statistically significant differences only among children who snored habitually when compared with those who did not snore. No discernible associations were identified between oSDB symptom scores and the volumes of subcortical structures, including the thalamus, basal ganglia, hippocampus, amygdala, or the cerebellum (Supplementary Table 3).

**Mediation analyses.** We used mediation analyses to determine whether and the extent to which the associations between oSDB factor score and behavioral measures were mediated by alterations in the regional cortical morphometric measures (Fig. 4a). We specifically focused on cortical volume as it represents the product of cortical thickness and surface area. Statistically significant regional mediation effects were identified for each of the CBCL scales (Fig. 4b), of which the strongest ten mediation effects were observed for regional gray matter volumes within the frontal lobe (Fig. 4c). For example, the relationship between the oSDB factor score and the CBCL attention problems were mediated by lower volumes of the precentral gyrus within each hemisphere (right; % mediated, 2.03 [0.74–3.51], $P < 0.001$; left; % mediated, 1.71; 95% confidence interval [CI], 0.91–2.84, $P < 0.001$). Similar effects were identified for the frequency of snoring as a symptom (Supplementary Fig. 2). Covariates for mediation models included age, sex, race/ethnicity, income, history of asthma as fixed effects and scanner ID as well as recruitment site as random effects.

To assess whether a clinical threshold exists for the frequency of symptoms associated with mediated CBCL measures, we explored the pairwise mediation effects derived from the frequency of snoring recategorized into clinically relevant categories of habitual (less than three nights a week) and non-habitual (three or more nights a week) snoring. Figure 4d compares the mediation effects attributable to the volume of the right precentral gyrus for the oSDB factor score, the frequency of snoring, non-habitual and habitual snoring, respectively. Children who snored habitually appeared to have statistically significant mediation effects derived from cortical volumetric alterations. However, the non-habitual snoring group was no different from the non-snoring group as its confidence intervals crossed zero for four CBCL scales with the greatest average mediation effects.

We additionally determined whether the mediation effects shown in Fig. 4 were also identified in the prediction of children with abnormal CBCL measures meeting the criteria for further

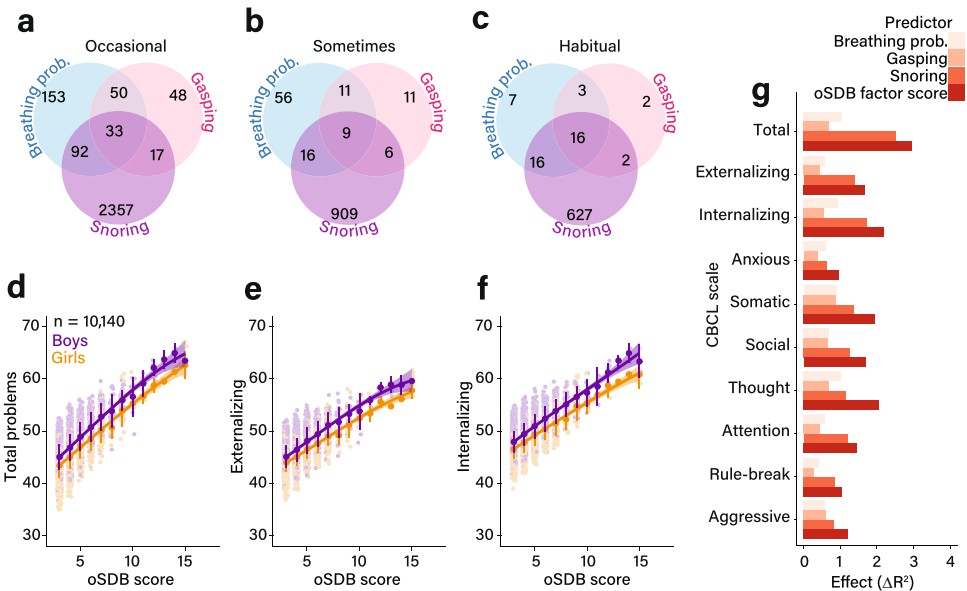

**Fig. 1 Relationship between symptom burden of obstructive sleep disordered breathing (oSDB) and Child Behavior Checklist scores. a–c** show the prevalence of parent-reported frequency of symptoms of obstructive sleep disordered breathing in the Adolescent Brain and Cognitive Development Study ($n = 10,140$). Venn diagrams show the number of children with each symptom in non-intersecting areas or in combination with other symptoms in the intersecting areas. The frequency of symptoms was categorized as occasional (**a**, once or twice a month), sometimes (**b**, once or twice a week) or habitual (**c**, >2 nights a week). The relationship between oSDB burden measured by the sleep-related breathing disorders subscale within the Sleep Disturbance Scale for Children and the three composite measures (total, externalizing, and internalizing problems) from the Child Behavior Checklist (CBCL) was assessed using generalized additive models. Models were adjusted for age, sex, race/ethnicity, the presence of asthma and total household income before taxes as fixed effects and study site as a random effect. **d–f** The relationship between the oSDB factor score and the predicted marginal values of three CBCL scales—total, externalizing, and internalizing problems, respectively. The adjusted distributions were fitted with a smoothing spline, with the error bars spanning one standard deviation around the mean predicted value of each composite CBCL measure with increasing oSDB score grouped by sex. **g** compares the effects of the parent-reported frequency of oSDB symptoms (e.g., snoring frequency) on problem behaviors by measuring the change in the overall proportion of variance in the generalized additive model following their addition to the null, covariates-only model. The null model comprised solely of fixed effect (age, sex, race/ethnicity, asthma, and total household income before taxes) and random effect (recruitment site) covariates. The comparisons of the base and covariate-adjusted models using analysis of variance yielded statistically significant results for all CBCL categories (two-sided $P < 10^{-16}$, unadjusted for multiple comparisons). The greatest effect was identified for the relationship between the oSDB symptom score and total problems. Among the individual symptoms, the frequency of snoring predicted CBCL scores better than others. Source data are provided as a Source Data file.

diagnostic evaluation. In Supplementary Fig. 3a, we show that the relative risk of abnormal CBCL scores was greater with habitual snoring compared to control children. This elevated risk was also identified for non-habitually snoring children albeit smaller in magnitude. The mediated effects for abnormal CBCL scores related to regional cortical volumes for the top ten cortical regions (Supplementary Fig. 3b, c) were greater in magnitude but regionally conserved compared to Fig. 4. For example, smaller volume of right precentral gyrus mediated the association between oSDB factor score and abnormal rule-breaking behavior (% mediated, 4.78 [1.55–10.48], $P < 0.001$). The frequency of snoring replicated this specific regional mediation effect (Supplementary Fig. 4, % mediated, 4.58 [1.67–11.44], $P < 0.001$).

## Discussion

In this national study of brain development in typically developing children aged 9–10 years, greater symptom burden of oSDB as reported by the parent was associated with higher scores measured by CBCL and thinner cortical gray matter within several frontal lobe regions. Importantly, the magnitude of the relationship between the oSDB symptoms and behavioral measures was mediated by regional cortical volumes, of which the strongest mediator was the volume of the right precentral gyrus. Together, these results provide evidence for brain-structure-related determinants of the relationship between parent-reported symptom burden of oSDB and problem behaviors.

The associations between asthma and race as identified by the parent with parent-reported oSDB scores mirror those from the previous studies of risk factors for oSDB in children[24]. The sex effect observed in the risk of oSDB in adults is potentially mitigated by weaker hormonal influence on the structure and function of the airway in prepubertal children[24].

Our study demonstrated stronger negative associations between oSDB symptoms and regional cortical thickness compared to surface area and volume. Neuroanatomic regions involved included the orbitofrontal sulci and gyri, the superior frontal sulci, parts of the insula and the occipitotemporal gyri. These findings are in general agreement with previous structural neuroimaging studies of children with obstructive sleep apnea[19,25]. As previously suggested[25], thinner cortices in these regions may indicate regional differences in susceptibility to the detrimental effects of sleep disruption with or without intermittent hypoxia observed in nocturnal upper airway obstruction. While causal effects cannot be inferred due to the cross-sectional nature of the current study, these results are consistent with a proposed mediational model that links oSDB with executive dysfunction via injury to the prefrontal cortex[17].

In our study, regional gray matter volumes mediated the relationship between oSDB symptoms and problem behavior scores. Brain regions included the precentral gyri, the supramarginal gyrus and the orbital and the superior frontal gyri. These regions are neural substrates for attention/working

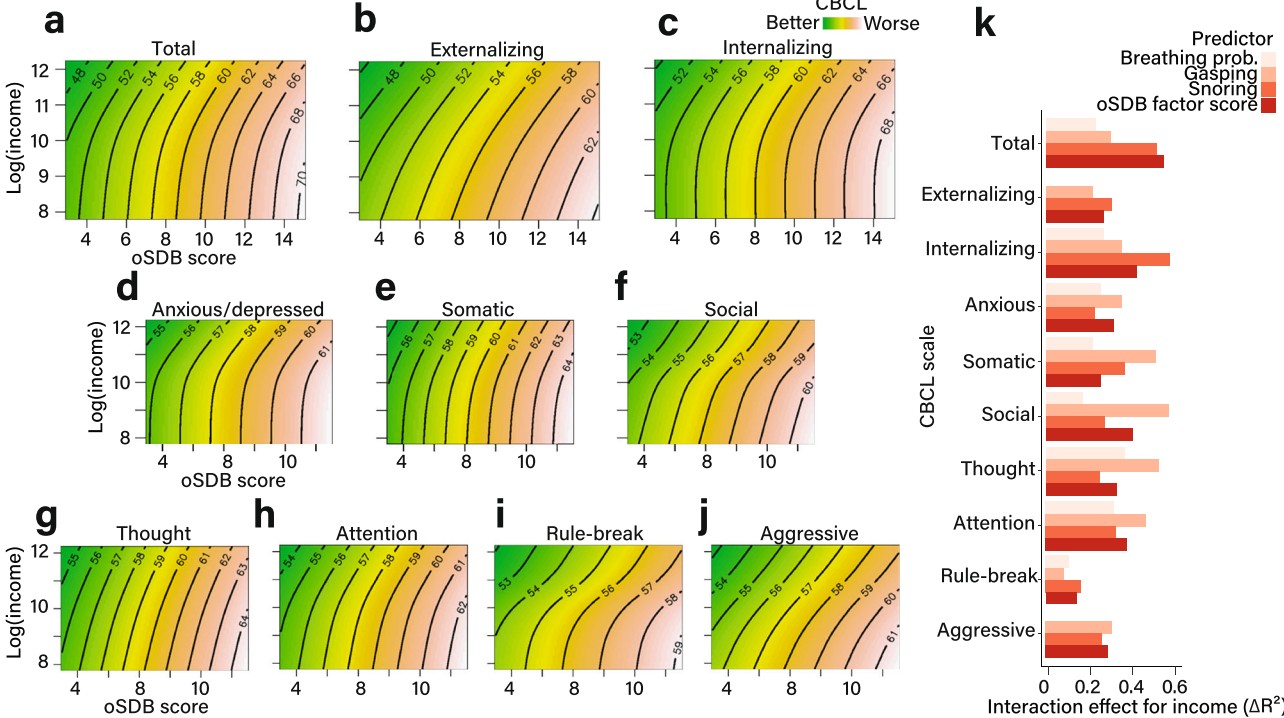

**Fig. 2 Interaction between household income and obstructive sleep disordered breathing in predicting problem behaviors. a–j** Each contour plot, derived from 10,140 children in the Adolescent Brain Cognitive Development study, shows the interaction between household income and obstructive sleep disordered breathing (oSDB) factor score in predicting the CBCL scores. The three composite CBCL scales are shown in **a–c** while the individual syndrome CBCL scales are shown in **d–j**. The median of each income class was log-transformed due to positive skew. The parent-reported obstructive sleep disordered breathing (oSDB) factor score is shown on the x-axis and the natural log of income on the y-axis. The angled orientation of the contours indicates an interaction between income and oSDB on each CBCL score. All models included the summed effects of age, sex, race/ethnicity, asthma as well as the recruitment site as a random effect. **k** The effect size estimates were calculated by measuring the change in the overall proportion of variance (expressed as a percentage) by adding the interaction term to the no-interaction model linking oSDB factor score to each CBCL measure. Source data are provided as a source data file.

memory, somatosensory, and phonological processing as well as behavioral inhibition[26–28]. The greatest mediation effect was identified for lower volumes of these regions accompanying the CBCL attention problems scale, which additionally validates the commonly observed attention problems in children with chronic upper airway obstruction[29]. Among each of the three key symptoms of oSDB, the frequency of snoring demonstrated the strongest association with higher CBCL scores in line with previous studies[7,9,30], as well as smaller volumes of bilateral precentral gyri supporting the use of the frequency of snoring to screen for oSDB[13]. Snoring is a highly sensitive but non-specific symptom of significant upper airway obstruction and habitual snoring currently merits further clinical evaluation according to various clinical guidelines[3,31]. Furthermore, the observed mediation effects for the relationship between the frequency of snoring and behavioral measures were more readily apparent in children with habitual snoring, supporting the threshold for screening as promulgated by the American Academy of Pediatrics[3]. However, some loss of statistical power may occur as a result of recategorization of Likert-like scales, and hence these results should also be viewed with caution. The evidence for a threshold effect for neuroanatomic changes directly or indirectly associated with the frequency of snoring may facilitate the refinement of screening practices to identify children with oSDB, and are at risk for neurobehavioral morbidity.

The associations of oSDB with cortical thickness as a total effect and volume as a mediation effect may be potentially explained by genetic and environmental influences on cortical surface area and thickness, that interact with each other in influencing regional gray matter volume[32].

Our study did not identify an association between oSDB burden and the volumes of subcortical structures. However, since disrupted microstructure and altered metabolites were reported in the hippocampus in children with obstructive sleep apnea[20], the most severe form of oSDB, these children may also have sub-cortical volume abnormalities.

Several design-related aspects of the current study enhanced its rigor and reproducibility. By using stratified sampling, the ABCD study design minimizes systematic errors[33]. Our study included greater representations of racial and socioeconomic subgroups compared to previous large datasets[6–8]. Surveys and behavioral assessments were performed in person and therefore mitigating the potential bias related to data collection by mail in some of the previous studies[7]. The assessment of behavior, while parent-reported, used the CBCL, a reliable, validated and structured scale compared to previous studies that focused on parent reports of hyperactivity and temper[7]. The use of centralized image processing and protocols to minimized between-site variation and improved the reliability of the MRI data. The response rate exceeds most community-based surveys in the past and the stratified sampling mitigated recruitment bias. The possible moderating influences related to socioeconomic status and contributions from recruitment setting, as well as influences related to the scanner technology were addressed by statistical models that controlled for these random factors.

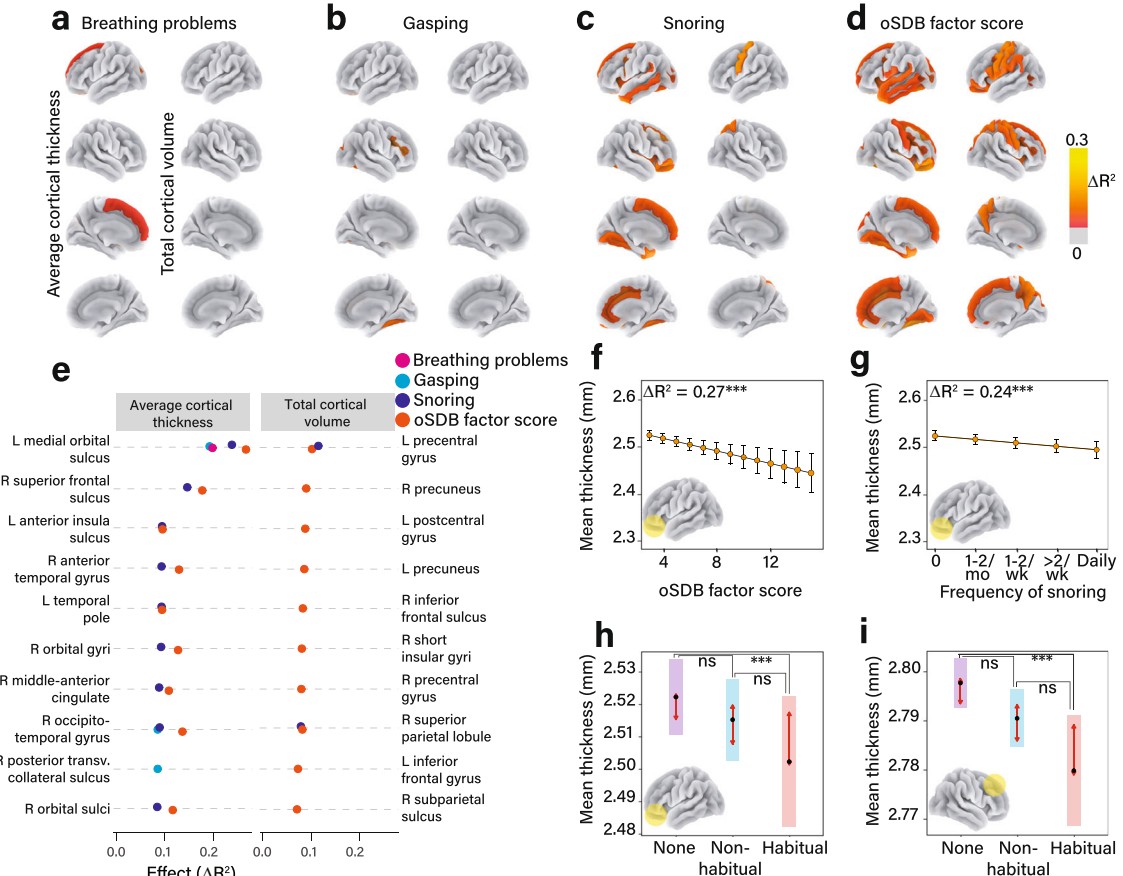

**Fig. 3 Obstructive sleep disordered breathing (oSDB) is associated with regional cortical thinning. a–d** Atlas-based effect size maps projected on a cortical surface demonstrate the relationship between the frequencies of individual symptoms, the total oSDB factor score and cortical morphometric variables (average cortical thickness (left) and total volume (right)) measured using magnetic resonance imaging (MRI). Among the individual symptoms, the greatest effect size was identified for the relationship between the frequency of snoring and average cortical thickness (**c**). The top ten cortical regions of interest (ROI) wherein the strongest effects were identified for each morphometric variable are shown in **e**. Effect sizes were calculated by measuring the change in overall proportion of variance by adding the predictor to the covariates-only model comprising age, sex, race/ethnicity, and household income as fixed effects and MRI scanner as a random effect. **f** The covariate-adjusted model for the relationship between oSDB factor score and the mean cortical thickness within the left medial orbital sulcus in 10,140 children. This relationship was replicated for the frequency of snoring as a predictor (**g**). The error bars in **f**, **g** span 95% confidence intervals around the estimated marginal mean cortical thickness. **h** shows a potential threshold associated with cortical thinning within the left medial orbital sulcus with pairwise comparisons between 661 children who snored habitually (>2 nights a week), and 6040 children who did not snore. However, there was no significant difference between 3439 children who snored non-habitually, defined as less than three nights a week, compared to the non-snoring children. **i** A similar comparison for the average thickness of the right superior frontal sulcus. A *P* value threshold of 0.05 was applied to all effect size maps following correction for false discovery. All pairwise comparisons in **h** and **i** were adjusted for multiplicity using the Tukey method with the arrows on either side of the estimated mean highlighting the regions of overlap of the estimated marginal means. These tests were two-sided. Cortical surface area was not associated with oSDB or any of the individual symptoms. Source data are provided as a source data file.

The principal drawback of the current study is its cross-sectional nature. Therefore, none of the analyses described in the manuscript assume directionality or causal aspects of these relationships due to the potential for bias in cross-sectional mediational models. Although our study aimed to partially mitigate this via statistical techniques, such as the use of false discovery control in the regional brain changes and bias-corrected bootstrap in the mediation models that reduce model uncertainty, this limitation cannot be fully addressed without longitudinal data[34]. The current study also recognizes potential biases related to parental reporting. For example, snoring, a key symptom of oSDB, can be assessed remotely by parents while respiratory pauses and nighttime breathing problems require in-person assessments. We also cannot rule out the potential for parental over-reporting of oSDB-related symptoms in children with behavioral problems. Furthermore, polysomnography was not used to quantify upper airway obstruction, which could have stratified the severity of oSDB, corroborated the parent-reported symptoms, and provided a more specific biological basis for these relationships by assessing the associated hypoxemia as well as the sleep disruption[3]. However, a few studies[13,35] also suggest that the frequency of oSDB symptoms should be independently examined when assessing children with sleep pathology, reinforcing its use in large community based studies such as the ABCD. More recently, others have also reported that children with and without polysomnographic evidence of oSDB are frequently indistinguishable regarding the amount, frequency, and the degree of sleep disturbance caused by symptoms of oSDB[36]. Therefore, cumulatively, parental assessments in the current study share the same advantages and disadvantages as previous national surveys from around the world in which parental reports of symptoms of oSDB were used without polysomnography for corroboration[6–8,10].

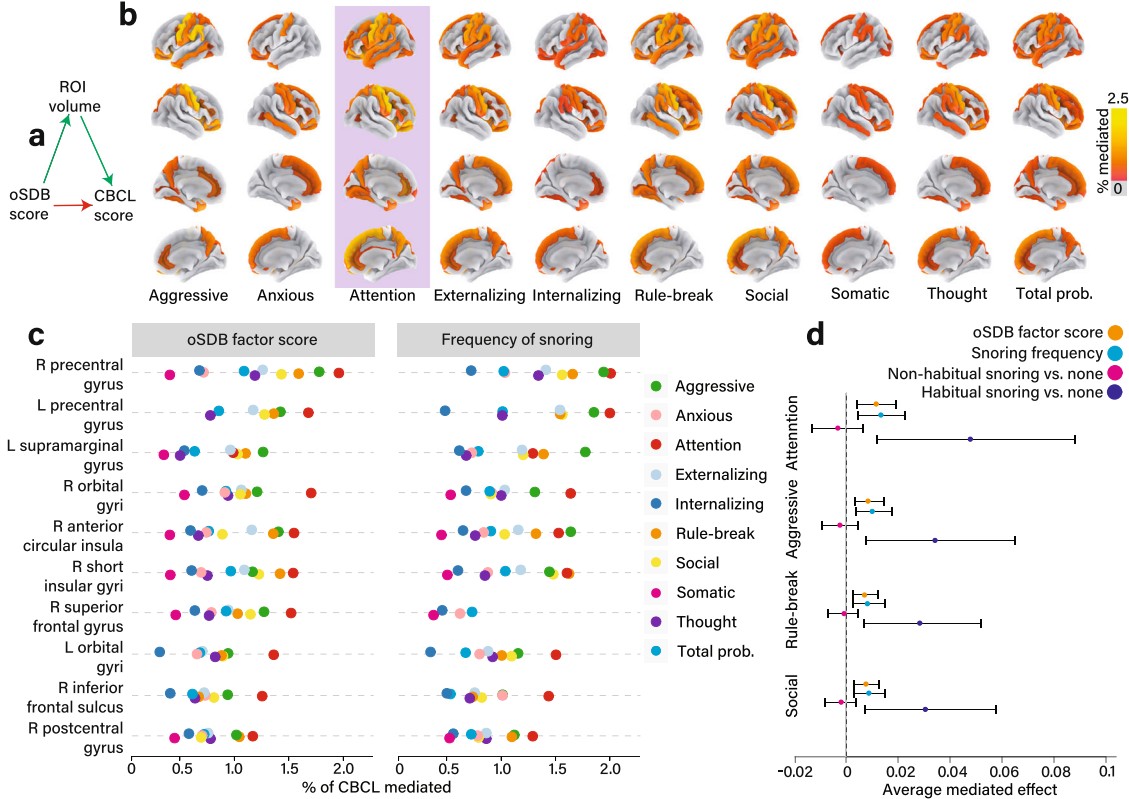

**Fig. 4 Frontal lobe regions mediate the relationship between obstructive sleep disordered breathing (oSDB) and Child Behavioral Checklist (CBCL) scores. a** General outline of the mediation model that estimates the extent of the covariate-adjusted relationship between oSDB factor score and the CBCL measures apportioned to alterations in cortical volume. These mediation effects expressed as a proportion of the total effect (% mediated) were projected on to atlas-based cortical effect size maps for CBCL scores (**b**). All mediation models included age, sex, race/ethnicity, history of asthma, and the total household income before taxes as fixed effects, and the recruitment site and the scanner serial number as random effects. The most widespread and strongest mediation effects were identified for attention problems (highlighted). **c** The top ten mediated effects for cortical regions of interest (ROI) for both oSDB factor score and the frequency of snoring showing similar effects. **d** Explores the clinical threshold for the frequency of snoring as a predictor for CBCL scores. Following recategorization of snoring frequency as none, non-habitual (less than three nights a week) and habitual (at least three nights a week), mediation effects were identified only for habitual snoring. All mediation effects were derived from jointly modeling two regressions—the first assessed the ROI using the predictor and the second assessed the CBCL score using the ROI as a predictor. The error bars span 95% confidence intervals associated with the average mediated effect and were obtained by bootstrapping 1000 replicates. A threshold of $P < 0.05$ following correction for false discovery was applied to all mediation models in addition to removing mediation effects whose confidence intervals crossed zero. Mediation effects for abnormal CBCL scores as a categorical variable are shown in Fig. S3. Source data are provided as a Source Data file.

Although asthma was included as a covariate in our analyses of the ABCD dataset, there may be other potential unmeasured confounders such as atopic pathology and parental snoring. Finally, the symptoms of upper airway obstruction used in the current study do not reliably differentiate between the various phenotypes of sleep disordered breathing defined by poly-somnography, ranging from primary snoring to obstructive sleep apnea. Our study points to a spectrum of brain structural changes accompanying oSDB and the behavioral morbidity; future focused studies can further delineate the nature of the insult, specifically the hypoxic changes, the sleep disruption, or possibly both[37].

The associations between oSDB and both structural[18,19,25,38] and functional brain alterations[20,39], have been reported specifically for brain regions within or close to the prefrontal cortex. The pattern of cortical changes in children with oSDB appears to overlap with imaging findings from adults in whom the affected areas include the insular, cingulate, and ventromedial prefrontal cortices[40]. More recently, cognitive deficits in children with sleep apnea were linked to disrupted white matter integrity in the dentate gyrus[41]. The majority of these studies included focused investigations of children from single center cohorts with a diagnosis of sleep apnea that included significantly smaller sample sizes[19,25,41]. As sleep apnea, a condition defined by poly-somnography, accounts for less than 10% of the 500,000 children with oSDB who undergo tonsillectomy and adenoidectomy annually in the United States[11], our study extrapolates the previous evidence to a larger group of children with oSDB that includes children with various severities of upper airway obstruction including sleep apnea. Furthermore, although it is well-known that socioeconomic variables such as household income and education are also associated with both brain structure[42] and oSDB[3], not all studies have controlled for these potential confounding factors[3]. Finally, all but one[18] of these studies focused solely on tests of general cognitive performance, despite the equal concern among children with chronic upper airway obstruction for impaired cognition and problem behaviors[3].

The strength of the findings described in this large and diverse study justifies and reinforces the early evaluation and screening of children with symptoms of upper airway obstruction. These results also provide population-based evidence for a biologically plausible model linking the burden of oSDB with neurobehavioral problems via brain characteristics shaped by hypoxia with or

without sleep disruption. While clinical guidelines from the various clinical societies[3,31,43,44] diverge in the optimal strategies for diagnosis and management of oSDB, there is no evidence to date to suggest the outcomes of one approach are decisively better than the other. Given that these differences in management strategies are shaped by the interactions between each of these specialties and children with oSDB, the current manuscript provides common ground to support the screening for and further evaluation of oSDB based on its observed associations with regional brain structural alterations. These results are also likely to spur longitudinal models, including that of other brain measures, such as functional connectivity, tissue integrity, and task-related functional imaging within the ABCD cohort.

## Methods

**Data source**. The ABCD study (https://abcdstudy.org) is a prospective, observational, national, 21-site assessment of brain development in children from ages 9–10 years through adulthood[45]. The rationale and design aspects of the study are well-described[33]. The ABCD study was approved by the central (University of California, San Diego) and the local institutional review boards at University of Maryland, Baltimore, University of Colorado, Boulder, University of Minnesota, the Laureate Institute for Brain Research, Oregon Health and Science University, University of Vermont, University of Pittsburgh, Virginia Commonwealth University, University of Rochester, University of Florida, Medical University of South Carolina, University of Michigan, University of Minnesota, University of Utah, SRI International, University of Wisconsin-Milwaukee, Children's Hospital of Los Angeles, Florida International University, Washington University in St. Louis, and Yale University. The current dataset (v2.0.1) includes 11,875 children enrolled by October 2018. Children aged 9–10 years along with their primary caregiver were recruited and informed consent obtained from the parent or guardian and assent obtained from children[46]. By its design, the ABCD study approximates the diversity of the U.S. population on sex, race/ethnicity, and socioeconomic status via stratified sampling[33]. Children were excluded if they had serious neurological or psychiatric diagnosis[47].

Dependent variables included assessments of problem behaviors and cortical morphometric variables from structural magnetic resonance imaging (MRI) of the brain. Participant age, biological sex at birth, and race/ethnicity were reported by the primary caregiver at enrolment. The body mass index (BMI) percentile score was calculated from the age-based and gender-based charts from the Centers for Disease Control[48]. The combined household income before taxes and the highest educational achievement of the primary caregiver were recorded. The demographic and anthropometric variables were included as covariates in all analyses.

**Assessment of oSDB**. The sleep disturbance scale for children is a 27-item clinically validated inventory described by six factor scores in children based on parent-reported symptoms during the preceding six months[49]. Two separate reviews of 21 and 57 pediatric sleep questionnaires[50,51] demonstrated that the Sleep Disturbance Scale in Children is the highest ranked scale within the domain for evidence-based assessment and instrument development. Widely used for studies of community screening and translated into multiple languages, the total and factor scores reliably differentiate clinical and control groups including children with oSDB[50,51]. Of the six subdomains, we utilized the sleep-related breathing disorder score that encompasses the frequencies of three key symptoms of oSDB—breathing difficulty during sleep, gasping or respiratory pauses and snoring. Although the parent scale is not validated against polysomnography, we used the key symptoms of oSDB in the subscale, which are identical or similar to those used in previous community-based surveys[7,8,10]. A five-point Likert-type score was obtained from the primary caregiver for each of these three items with response options of 1 = never, 2 = occasional (once or twice a month), 3 = sometimes (once or twice a week), 4 = habitual (more than twice a week but not daily), and 5 = daily. In line with the role of symptom burden in screening for oSDB in a clinical setting[3,13] and as recommended by the original validation study[49], we assessed the factor score by adding the individual symptom scores (range: 3–15) with higher scores indicating greater burden of oSDB. We also assessed individual symptoms as independent predictors in addition to the factor score. Exploratory recategorization of the frequency of snoring was performed for additional comparisons, for which snoring at least three nights a week was considered habitual and less frequent snoring was non-habitual[3].

**Assessment of problem behaviors**. Problem behaviors in the ABCD cohort were assessed using parent responses to the Child Behavior Checklist (CBCL), a validated and widely-used assessment of childhood behavior spanning emotional, social, and behavioral domains[52]. The following syndrome scales were obtained for the current study: anxious/depressed, somatic complaints, social problems, thought problems, attention problems, rule-breaking behavior, aggressive behavior, in

addition to the broadband scales of internalizing, externalizing and total problems[53]. Raw scores were converted to $t$-scores using gender-based and age-based norms from population-based studies, with higher scores indicating more severe behavior problems. For additionally exploratory analysis, we also used published cut-offs for children needing further evaluation[23].

**Structural brain imaging**. All children underwent structural brain MRI according to standardized protocols. Locally acquired T1-weighted images were processed at the Data Analysis, Informatics and Resource Center (DAIRC) of the ABCD study[54,55]. Images from the scans were processed using Freesurfer v5.3.0 (https://surfer.nmr.mgh.harvard.edu), projected onto a spherical atlas, generating 74 cortical parcellations within each cerebral hemisphere. Morphometric variables included the average cortical thickness, surface area, and volume of each region and the whole brain. Volumetric reconstructions of subcortical structures were also obtained. A detailed description of the imaging protocols is provided in the Supplementary Methods.

**Statistical analysis**. The relationship between the predictor variables and each CBCL measure was assessed using generalized additive modeling (Supplementary Methods). This method, pre-specified in the study protocol, facilitates the non-linear modeling of socioeconomic status, a well-described confounder in studies of the relationships between snoring and neurobehavioral outcomes.[3] Missing data, which is known to be low in the ABCD study, was excluded. The scanner serial number and the study site were included as random factors. The household income was log-transformed due to positive skew. Caregiver educational status was converted to approximate the number of years of education. Effect sizes were obtained for each model as the large sample size of the ABCD study derives low $P$ values for even small effect sizes. Therefore, the relationship between the oSDB symptom and factor scores, and each CBCL measure was calculated by determining the change in the overall proportion of variance (adjusted $R^2$) of the predicted model expressed as a percentage. Sex-based subgroup effects were determined by comparison of the effects for boys and girls. Models including the predictive term with or without interactions with income were compared to the covariate-only null model utilizing a likelihood ratio test. Only parsimonious models that comprised statistically significant variables that improved the overall fit were included. Similarly, the relationship between overall and individual oSDB symptom scores and each cortical region of interest was assessed. The effect size maps were subsequently projected on a cortical surface incorporating the threshold for statistical significance after adjustment for false discovery. Demographic and anthropometric variables were included as covariates in all models.

Next, the association of oSDB symptom scores with each CBCL measure attributable to regional morphometric characteristics was calculated using mediation analysis by concurrently fitting two covariate-adjusted regression models[56]. The first model assessed the prediction of each morphometric variable by the oSDB symptom score. The second model determined the prediction of CBCL measure by the oSDB symptom score. The jointly modelled mediation effect from both regressions was projected onto a cortical surface map after adjustment for false discovery. Given the potential for bias related to the application of mediation models using cross-sectional data[34], we used a more rigorous approach towards estimation of model uncertainty using the bias-corrected bootstrap for univariate model[57].

The generalized additive models for regression analyses were created using R v3.6.1 (https://cran.r-project.org) using the mgcv package. Mediation analysis was carried out using the mediation package (https://cran.r-project.org/web/packages/mediation). All effect size maps for the brain measures were generated using the Python-based nilearn package v0.5.2 (https://nilearn.github.io/). Due to the multiplicity associated with statistical testing, $P$ values were corrected for false discovery, with corrected $P$ values below 0.05 considered statistically significant.

**Reporting summary**. Further information on research design is available in the Nature Research Reporting Summary linked to this article.

## Data availability
The ABCD data repository grows and changes over time. The ABCD data that support the findings of this study are available in National Institutes of Mental Health Data Archive (NDA) with the digital object identifier (DOI) 10.15154/1520518[58]. Source data are provided with this paper.

## Code availability
The R code pertaining to the figures in this manuscript is provided at https://github.com/sjoh2574/oSDB. Custom code used in this manuscript can be obtained upon request from the authors.

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

## Acknowledgements

Data used in the preparation of this article were obtained from the Adolescent Brain Cognitive Development<sup>SM</sup> (ABCD) Study (https://abcdstudy.org), held in the NIMH Data Archive (NDA). This is a multisite, longitudinal study designed to recruit more than 10,000 children age 9–10 and follow them over 10 years into early adulthood. The ABCD Study® is supported by the National Institutes of Health and additional federal partners under award numbers U01DA041048, U01DA050989, U01DA051016, U01DA041022, U01DA051018, U01DA051037, U01DA050987, U01DA041174,

U01DA041106, U01DA041117, U01DA041028, U01DA041134, U01DA050988, U01DA051039, U01DA041156, U01DA041025, U01DA041120, U01DA051038, U01DA041148, U01DA041093, U01DA041089, U24DA041123, and U24DA041147. A full list of supporters is available at https://abcdstudy.org/federal-partners.html. A listing of participating sites and a complete listing of the study investigators can be found at https://abcdstudy.org/consortium_members/. ABCD consortium investigators designed and implemented the study and/or provided data but did not necessarily participate in analysis or writing of this report. This manuscript reflects the views of the authors and may not reflect the opinions or views of the NIH or ABCD consortium investigators.

## Author contributions

Concept and design: Isaiah, Ernst, Cloak and Chang; Acquisition, analysis, or interpretation of data: All authors; Drafting of the manuscript: Isaiah; Critical revision of the manuscript for important intellectual content: All authors; Statistical analysis: Isaiah; Obtained funding: Ernst, Clark and Chang.

## Competing interests

Amal Isaiah has patents (pending or granted) related to the diagnosis and treatment of sleep apnea in adults using ultrasound. These have been licensed by the University of Maryland, Baltimore. They are not discussed in the current manuscript. All others declare no competing interests.
