## [Peer Review File · Nature Communications]

Reviewer #1 (Remarks to the Author):

Sleep disordered breathing is very common in children and ranges in severity from primary snoring which is not associated with sleep disruption or gas exchange abnormalities to obstructive sleep apnoea. All severities of sleep disordered breathing have been associated with adverse effects on behaviour and neurocognition. Previous studies that have examined changes to brain structure have been limited to small sample sizes – due primarily to the cost of MRI. This is the largest study to date (including over 10,000 children) that has related sleep disordered breathing and behaviour with measures of brain structure assessed by MRI. The study found that greater severity of sleep disordered breathing was related to worse behaviour and that volumes of multiple cortical regions within the frontal lobe mediated the relationship between severity of sleep disordered breathing and behaviour outcomes. The manuscript has been well written and the data have been comprehensively analysed. The limitations of the cross-sectional nature of the study and that parental report of sleep disordered breathing severity rather than the gold standard of polysomnography study and parental report of behaviour are acknowledged. The study provides important new information which supports assessment and treatment of sleep disordered breathing in children.

Minor comments:

A mixture of English and American spelling is used throughout the manuscript. If English spelling is to be used z needs to be replaced with s in such words as recognises, characterised etc and an a to hypoxaemia.

Consider splitting the last sentence of the abstract.

Reviewer #2 (Remarks to the Author):

Structural alterations in the frontal lobe mediate the impact of snoring and associated symptoms on childhood behaviour

Thank you for inviting me to review this very interesting study, the goal of which was to evaluate the relationship between snoring, childhood behaviours and changes within the cortical grey matter.

Overall, this is a well written manuscript which describes the relationship between snoring, childhood behaviours and changes within the cortical grey matter. However, what was not very clear was the novelty of these data and how clinically significant/relevant these results are. For example, the overall CBCL results are within clinically normal range - what would the clinical threshold for concern for cortical changes? And threshold for intervention to attempt to reverse changes. This discussion is essential to provide further context and interpretation of these results.

Abstract:

What is meant by reversal of the associated morbidity (last line)- of the brain or behaviour or both (line 32, page 2)?

Introduction:

1. oSDB- this can occur without pauses in breathing, specifically obstructive hypopneas are associated with reduced airflow rather than pauses.
2. Please clarify definition of adverse behaviours at the outset, line 44 page 2.
3. There have been other studies that have evaluated SDB, brain imaging and behaviours. Novelty of this dataset is not clear in the introduction. Please also provide a hypothesis for your study.

Methods

1. My concern is that the symptoms of oSDB and SDB burden may have been over-estimated by the questionnaires. How many of these children had asthma and/or allergy? Asthma is an important confounder and can commonly co-exist with OSA and nocturnal sounds of OSA and asthma are difficult to disentangle. This may be particularly as the peak age of OSA tends to be 8 years or less (this cohort is older at age 10 years) and the prevalence of symptoms is high at >40% in an otherwise healthy cohort. Similarly, allergies may be associated with allergic rhinitis predisposing to infrequent snoring.
2. Was family/parental history of snoring recorded and controlled for?

3. What is a clinically significant cut off for CBCL scores?
4. Did any of these children have clinical concerns of behavioural problems?

Results

While I appreciate all the tables and figures as well as their quality, it was not ideal to navigate back and forth between the main PDF and supplementary figures and tables. Could the figures and tables be further streamlined?

1. Please provide context around CBCL normative data vs these results. The mean scores appear to be within normal range for every 'adverse behaviour'.
2. How many of this population had abnormal CBCL scores and how do they compare to no SDB populations?
3. Please direct the reader whether a CBCL score increase of 3.3 units clinically significant?
4. Analyses should be repeated using habitual snorers vs sometimes vs occasional (or present snoring on a scale 1-5 as per your Likert scale for data collection) to see if data analyses are replicated when compared to using the composite score. Specifically, it would be helpful to see if 'snoring more' has a dose dependent relationship with CBCL and cortical changes.
5. I did not appreciate Fig 4d demonstrated the effect size for frequency of snoring and average cortical thickness rather it was the presence of snoring per se- is that correct?
6. Would suggest also doing mediation models using habitual snoring only rather than oSDB factor score. That is, evaluate the strength of the relationship between snoring as an independent variable and Cortical morphometry and CBCL as dependent variable (as per Fig 5 data).
7. The magnitude (not statistical magnitude) of changes in the cortex, either thickness or volume should be outlined more clearly. Specifically, are these changes clinically relevant?

Discussion

1. The discussion highlights well the strengths particularly the large longitudinal dataset especially the collection of MRI data in > 10,000 children.
2. While these data are interesting, and considered in the context of other studies, the discussion does not really highlight novelty of these data compared with other studies.
3. Asthma and/or allergy data – if not available, then the lack of this data should be added to limitations.
4. Line 267 onwards in discussion...'the current supports novel common ground to support screening for the condition based on its observations with regional brain structure'.... please clarify this statement

Reviewer #3 (Remarks to the Author):

By analyzing data from a very large cohort of preadolescents (n=10,140), the authors reported strong associations between obstructive sleep disordered breathing (oSDB) symptoms and regional cortical thickness in the brain. Since oSDB is commonly related to neurobehavioral deficits, the findings in this study can help connect brain structural changes to both oSDB and the associated neurobehavioral abnormalities. This novel study is expected to have great interest to the scientific community. The reported results are well supported by sound statistical analyses and the methods are generally rigorous. The study leveraged a large, multi-center imaging database from the ABCD project, comprising not only structural MRI but also functional and diffusion MRI from which brain connectivity information can be derived. It is somewhat puzzling why none of the brain connectivity results, which could be relevant to brain alterations associated with oSDB, was mentioned in the manuscript. Inclusion of brain connectivity information can potentially enhance the conclusions, provided that it is supportive. Another weakness is that the casual aspects of the relationships described in the manuscript are uncertain. A longitudinal study design, as opposed to a cross-sectional design employed in the study, may help provide clarification. Overall, this is a very interesting study with potentially high impact. However, the limitations outlined above, together with additional limitations that the authors discussed in the manuscript, somewhat weakened the conclusions.

REVIEWER COMMENTS

Reviewer #1 (Remarks to the Author):

Sleep disordered breathing is very common in children and ranges in severity from primary snoring which is not associated with sleep disruption or gas exchange abnormalities to obstructive sleep apnoea. All severities of sleep disordered breathing have been associated with adverse effects on behaviour and neurocognition. Previous studies that have examined changes to brain structure have been limited to small sample sizes – due primarily to the cost of MRI. This is the largest study to date (including over 10,000 children) that has related sleep disordered breathing and behaviour with measures of brain structure assessed by MRI. The study found that greater severity of sleep disordered breathing was related to worse behaviour and that volumes of multiple cortical regions within the frontal lobe mediated the relationship between severity of sleep disordered breathing and behaviour outcomes. The manuscript has been well written and the data have been comprehensively analysed. The limitations of the cross-sectional nature of the study and that parental report of sleep disordered breathing severity rather than the gold standard of polysomnography study and parental report of behaviour are acknowledged. The study provides important new information which supports assessment and treatment of sleep disordered breathing in children.

We are pleased to note the reviewer's positive comments.

Minor comments:

A mixture of English and American spelling is used throughout the manuscript. If English spelling is to be used z needs to be replaced with s in such words as recognises, characterised etc and an a to hypoxaemia.

We have corrected and used only American spelling in the revised manuscript. In addition, we have made a number of changes throughout the manuscript to more accurately reflect the appropriate terminology. These include, “behavioral measures, scales, or scores” instead of “behavioral outcomes” to avoid causality, as well as using “problem behaviors”, a more specific usage compared to the general term “behavior”.

Consider splitting the last sentence of the abstract.

We have split the sentence to enhance its readability, “...These results provide population-level evidence for regional structural alterations in cortical gray matter accompanying problem behaviors in children with oSDB. Timely recognition and treatment of oSDB may ameliorate these changes and the associated neurobehavioral morbidity while the frontal lobe still retains age-dependent plasticity.....”

Reviewer #2 (Remarks to the Author):

Structural alterations in the frontal lobe mediate the impact of snoring and associated symptoms on childhood behaviour

Thank you for inviting me to review this very interesting study, the goal of which was to evaluate the relationship between snoring, childhood behaviours and changes within the cortical grey matter.

Overall, this is a well written manuscript which describes the relationship between snoring, childhood behaviours and changes within the cortical grey matter. However, what was not very clear was the novelty of these data and how clinically significant/relevant these results are. For example, the overall CBCL results are within clinically normal range - what would the clinical threshold for concern for cortical changes? And threshold for intervention to attempt to reverse changes. This discussion is essential to provide further context and interpretation of these results.

We thank the reviewer for the overall comments and appreciate the feedback. We have addressed these overall comments and the specific comments below separately.

“...However, what was not very clear was the novelty of these data and how clinically significant/relevant these results are...”

Globally, numerous population-based studies continue to highlight the relationship between children with symptoms of oSDB and behavioral outcomes. These studies have focused on symptoms of oSDB without polysomnographic assessments due to the constraints of using polysomnography for such large populations. A typical example is “Prevalence and Risk Factors of Habitual Snoring in Primary School Children” published in *Chest* in 2010. This study by Li et al. used postal surveys to query the relationship between habitual snoring and parent-reported behavior in ~9,000 children. Numerous others have replicated this relationship. In fact, even when polysomnography has been employed, parent-reported frequency of snoring has been shown to be a better predictor of both problem behaviors and cognitive deficits in children (e.g. *Smith DL, Gozal D, Hunter SJ, Kheirandish-Gozal L. Frequency of snoring, rather than apnea-hypopnea index, predicts both cognitive and behavioral problems in young children. Sleep medicine. 2017 Jun 1;34:170-8.* However, to date, there has been no study that examined the neuroanatomic basis for the behavioral problems in population-based investigations of oSDB in children. Horne et al. (*Horne RS, Roy B, Walter LM, Biggs SN, Tamanyan K, Weichard A, Nixon GM, Davey MJ, Ditchfield M, Harper RM, Kumar R. Regional brain tissue changes and associations with disease severity in children with sleep-disordered breathing. Sleep. 2018 Feb;41(2):zxx203*) remains the only work to date that investigated problem behaviors and neural injury in children using DTI to measure whole brain mean diffusion in oSDB. However, this study (n=18 with SDB and n = 20 controls) could not, given the small sample size, include the mediational model that utilizes a nationally representative large dataset we describe in our paper with 10,000 children that was obtained following rigorous sampling efforts. A mediational model would require explicit joint modeling of two regression equations, i.e. ROI on oSDB and CBCL on ROI after incorporating the relevant covariates.

Finally, we wish to highlight the major contribution of confounders. This well-known problem was highlighted by a publication by Celle et al. (*Celle S, Delon-Martin C, Roche F, Barthelemy JC, Pépin JL, Dojat M. Desperately seeking grey matter volume changes in sleep apnea: a methodological review of magnetic resonance brain voxel-based morphometry studies. Sleep medicine reviews. 2016 Feb 1;25:112-20.*). We have cited a portion of their abstract.

“...Finally, based on strict methodological criteria, only three studies reported robust, but conflicting, results. No clear evidence has emerged and exploring brain alteration due to obstructive sleep apnea should thus be considered as an open field. We provide recommendations for designing additional robust voxel-based morphometry studies, notably the use of larger cohorts, which is the only way to solve the underpowered issue and the underestimated role of confounders in neuroimaging studies...”

This citation has been added to our revision.

As we explain further in the manuscript (**page 3, para 2**), “...However, ascertaining this poorly understood link between oSDB and **problem behavior** in previous population-based studies requires the interrogation of brain-related changes common to both¹⁶. Evidence of brain structural changes in children with oSDB **have been** limited to a few small studies of associations between obstructive sleep **apnea** and **alterations in brain tissue or neuronal integrity**¹⁷⁻¹⁹. The changes within the cortical gray matter defining the relationship between oSDB and **behavioral measures** in large population-based cohorts remain unverified. Validating this **concurrent** negative relationship also requires robust statistical control for confounding from demographic and socioeconomic factors that **often** impact the generalizability of small studies^{20,21} ...”

In response to the reviewer’s comment, we have added the following paragraph to further highlight the novelty of the study (**page 3, para 2, line 68 onwards**): “...**Cumulatively, the results from previous studies highlight two major facets of the gaps in our knowledge derived from large investigations of behavior in children with oSDB. First, no population-based neuroimaging study has investigated whether snoring in children is associated with underlying neuroanatomic alterations following statistical control for common confounders such as socioeconomic status. Second, no study, large or small, has investigated whether the association between oSDB and behavioral measures in children could be attributable to structural alterations in the brain. Addressing these gaps could streamline the evaluation and management of children**

with symptoms of oSDB by better delineating the pathophysiology of oSDB-related neurobehavioral morbidity such as problem behaviors and poor school performance, identifying children at risk for the morbidity based on symptoms and enabling clinicians to monitor clinically significant post-treatment recurrence of oSDB in children...”

“...For example, the overall CBCL results are within clinically normal range - what would the clinical threshold for concern for cortical changes? And threshold for intervention to attempt to reverse changes. This discussion is essential to provide further context and interpretation of these results...”

In this revision, we have focused on improving the clinical relevance of these findings by (i) examining a clinical threshold effect for the negative association between the frequency of snoring and thinning of the cortex, and (ii) additional comprehensive mediation analysis for abnormal CBCL measures.

We added the following panels to **Figure 3** for further illustrating the threshold effect for clinical screening. Here we specifically focused on the thinning observed in the left medial orbital sulcus (**f-h**) and the right superior frontal sulcus – both of which are part of the frontal lobe and may be implicated in problem behaviors.

Excerpt from **Figure 3** legend: (**f**) shows the covariate-adjusted model for the relationship between oSDB factor score and the mean cortical thickness within the left medial orbital sulcus. This relationship was replicated for the frequency of snoring as a predictor (**g**). (**h**) shows a potential clinical threshold for cortical thinning within the left medial orbital sulcus with pairwise comparisons among children who snored habitually, i.e. at least three nights a week, compared to children who did not snore. However, there was no significant difference between children who snored non-habitually, defined as less than three nights a week, compared to non-snoring children. (**i**) shows a similar comparison for the average thickness of the right superior frontal sulcus. A P value threshold of 0.05 was applied to all effect size maps following correction for false discovery. All pairwise comparisons in were adjusted for multiplicity using the Tukey method with the arrows on either side of the estimated mean highlighting the regions of overlap of the estimated marginal means. Cortical surface area demonstrated no association with oSDB or any of the individual symptoms....”

Furthermore, we added the following panel to **Figure 4**, wherein we show a similar threshold effect for the mediation effect of changes in the volumes of the right precentral gyrus defining the relationship between oSDB factor score, the frequency of snoring and the pairwise comparisons of children with snoring categorized as non-habitual or habitual with those without history of snoring. Here we show that the only confidence interval that crossed a zero was related to the bootstrapped mediation effects related to the predictor non-habitual snoring (non-snoring children were controls). These findings show once again that the morphometric alterations that accompany the relationship between snoring and adverse behavioral measures may become apparent only above this threshold.

We also added a section for the clinical translation of these findings in **Page 10, para 1 as previously mentioned**: “...Therefore, these results, derived from simultaneously modeled effects of oSDB on structural alterations in the

brain as well as behavioral scores, provide a novel neuroanatomic basis for commonly reported behavioral problems in children with symptoms of oSDB. Furthermore, the observed mediation effects for the relationship between the frequency of snoring and behavioral measures were more readily apparent in children with habitual snoring, supporting the threshold for screening as promulgated by the American Academy of Pediatrics³. However, some loss of statistical power may occur as a result of recategorization of Likert-like scales and hence these results should also be viewed with caution. The evidence for a threshold effect for neuroanatomic changes directly or indirectly attributable to the frequency of snoring may facilitate the refinement of screening practices to identify children with oSDB and are at risk for neurobehavioral morbidity...”

Abstract:

What is meant by reversal of the associated morbidity (last line)- of the brain or behaviour or both (line 32, page 2)?

We clarified our reference to both outcomes in the abstract, “...These results provide population-level evidence for regional structural alterations in cortical gray matter accompanying problem behaviors in children with oSDB. Timely recognition and treatment of oSDB may ameliorate these changes and the associated neurobehavioral morbidity while the frontal lobe still retains age-dependent plasticity...”

Introduction:

1. oSDB- this can occur without pauses in breathing, specifically obstructive hypopneas are associated with reduced airflow rather than pauses.

We agree and therefore modified the sentence (page 2, para 2) to better reflect the definition of oSDB, “...Obstructive sleep disordered breathing (oSDB) in children is characterized by resistance to airflow in the upper airway, resulting in snoring and difficulty breathing with or without respiratory pauses during sleep¹...”

2. Please clarify definition of adverse behaviours at the outset, line 44 page 2.

We have modified this sentence to include the definition (page 2, para 3): “...The importance of parent-reported symptoms to screen for oSDB is highlighted by large, national, population-based studies demonstrating a strong bidirectional relationship between oSDB symptoms and problem behaviors such as inattention, hyperactivity and aggression...”

3. There have been other studies that have evaluated SDB, brain imaging and behaviours. Novelty of this dataset is not clear in the introduction. Please also provide a hypothesis for your study.

In an earlier response to the reviewer’s concern recording the novelty of the study, we have strived to provide clarity to the importance of the findings in the revised version.

We restate the addition to page 4, para 1, line 68 onwards here: “Cumulatively, the results from previous studies highlight two major facets of the gaps in our knowledge derived from large investigations of behavior in children with oSDB. First, no population-based neuroimaging study has investigated whether snoring in children is associated with underlying neuroanatomic alterations following statistical control for common confounders such as socioeconomic status. Second, no study, large or small, has investigated whether the association between oSDB and behavioral measures in children could be attributable to structural alterations in the brain. Addressing these gaps could streamline the evaluation and management of children with symptoms of oSDB by better delineating the pathophysiology of oSDB-related

neurobehavioral morbidity such as problem behaviors and poor school performance, identifying children at risk for the morbidity based on symptoms and enabling clinicians to monitor clinically significant post-treatment recurrence of oSDB in children...”

Page 4, para 2, line 85 onwards now explicitly states the hypothesis, “...We **hypothesized** that the association between parent reported symptoms of oSDB and **behavioral** outcomes in children is attributable to alterations in brain structure including regional cortical thickness, surface area and volume...”

Methods

1. My concern is that the symptoms of oSDB and SDB burden may have been over-estimated by the questionnaires. How many of these children had asthma and/or allergy? Asthma is an important confounder and can commonly co-exist with OSA and nocturnal sounds of OSA and asthma are difficult to disentangle. This may be particularly as the peak age of OSA tends to be 8 years or less (this cohort is older at age 10 years) and the prevalence of symptoms is high at >40% in an otherwise healthy cohort. Similarly, allergies may be associated with allergic rhinitis predisposing to infrequent snoring.

We appreciate the reviewer’s comments. We realize that stating the prevalence of any symptom of oSDB may be misleading and therefore removed it. Instead, we used the definition of habitual snoring as described in the technical report by Marcus et al. (Marcus CL, Brooks LJ, Ward SD, Draper KA, Gozal D, Halbower AC, Jones J, Lehmann C, Schechter MS, Sheldon S, Shiffman RN. Diagnosis and management of childhood obstructive sleep apnea syndrome. *Pediatrics*. 2012 Sep 1;130(3):e714-55.) who reviewed 25 studies of habitual snoring with an estimated pooled prevalence of the condition as being 3-12%. Consequently, the prevalence of habitual snoring in the ABCD study matches this range at 6.5%. Furthermore, we have appended the prevalence of these clinically relevant categories of snoring (none, non-habitual and habitual) to Table 1 and added this to the text as well (**page 4, para 1, line 94**), “...**Habitual snoring, defined as snoring at least three nights a week, was reported in 661 children (6.5%). In addition, snoring was the most predominant symptom of oSDB reported by the parents (Fig. 1a-c). A total of 1,726 (17.0%) children had a diagnosis of asthma...**”

Allergies are reported in a non-specific manner in the ABCD dataset as it encompassed food allergies, a common condition in this age group. We agree with the reviewer that allergies may co-exist with oSDB. Due to this missing variable, we have added a statement in the limitations (**page 13, para 1, line 292**), “...**Although asthma was included as a covariate in our analyses of the ABCD dataset, there may be other potential unmeasured confounders such as atopic pathology and parental snoring...**”. Here we used the term *atopic* to refer to any upper airway inflammatory condition.

Pursuant to the reviewer’s recommendation, we re-analyzed all behavioral GAM models and the mediation models wherein asthma was identified as a confounder. While there were no major departures from the previous results, the addition of asthma as a statistically significant covariate improved model fits and reduced some of the effect sizes of oSDB slightly. We thank the reviewer for facilitating the identification of more accurate models.

2. Was family/parental history of snoring recorded and controlled for?

No, once again, this has been listed as a potential limitation (**page 13, para 1, line 292**), “...**Although asthma was included as a covariate in our analyses of the ABCD dataset, there may be other potential unmeasured confounders such as atopic pathology and parental snoring...**”

3. What is a clinically significant cut off for CBCL scores?

We used the recommended cutoffs from Pandolfi et al. (Pandolfi V, Magyar CI, Dill CA. An initial psychometric evaluation of the CBCL 6–18 in a sample of youth with autism spectrum disorders. *Research in Autism Spectrum Disorders*. 2012 Jan 1;6(1):96-108.). These cutoffs have been incorporated into the descriptions of the CBCL scores in **Table S1** and n with % provided for each category. The legend has been modified, “...**Clinical t-scores are ≥ 64 for**

internalizing and externalizing problems, while borderline scores are 60-63 for these scales. For all other scales except those indicated by NA for which cut-offs have not been defined, clinical t-scores are ≥ 70 and borderline for 65-69. Children meeting the criteria for borderline scores and above are deemed candidates for further evaluation...”

4. Did any of these children have clinical concerns of behavioural problems?

We have now added the n and % for children with clinically significant CBCL scores.

In the revised version, we have taken due consideration of the reviewer’s concerns regarding the importance of clinically significant CBCL scores as well as other threshold-defined variables that provide additional interpretability in a clinical setting. Therefore, we repeated the mediation analysis for children who merited, at least by CBCL criteria, further diagnostic evaluation. These results, which largely replicate the results of **Figure 4**, are now presented in **Figure S3**. In addition, we show that the risk of abnormal behavior is increased with habitual snoring (**panel a**).

Results

While I appreciate all the tables and figures as well as their quality, it was not ideal to navigate back and forth between the main PDF and supplementary figures and tables. Could the figures and tables be further streamlined?

We thank the reviewer for the positive comments.

In order to streamline the figures and their presentation, we modified every figure and merged many panels. Other modifications include re-analyzing the data by following the reviewer’s suggestion to consider the history of asthma as a covariate in models that included CBCL as an outcome. We also focused on statistical parsimony, i.e., we removed covariates that had no meaningful improvements to the model fit. This results in the retention of household income and removal of parental education from models wherein they were simultaneously added. This resulted in an increase in overall R^2 values.

A summary of changes to the figures is provided below:

Figure 1 comprises panels merged from previous **Figures 1, 2 and 3**. The lack of relationship between oSDB and overall cortical morphometric measures was retained as text.

We believe the interaction between household income and oSDB in predicting the CBCL outcome scores is important and, therefore, we show these results in **Figure 2** along with the magnitude of the interactions with individual oSDB symptoms.

Figure 3 now incorporates the effect of the frequency of snoring on regional cortical thickness further explored in panels **f-i**. Here we show that the statistically significant pairwise comparisons with control children were observed among children with history of habitual snoring, but not in those with non-habitual snoring. These panels now address the reviewer’s comments about the contextual relevance of these findings by adding the threshold of snoring frequency at which brain injury may be observable.

Figure 4 now includes information merged from a previous supplemental figure indicating the mediation effects for all outcomes. In addition, we added a new panel (**d**) that shows that the mediation effects for regional cortical volume were not identified for non-habitual snoring. These results, along with the panels (**f-i**) in **Figure 3** provide important new information to demonstrate a threshold effect for the frequency of snoring.

The new **Figure S2** was added following the reviewer's recommendation to assess whether the mediation effects observed in **Figure 4** were replicated for the frequency of snoring.

A new **Figure S3** contains novel mediation analysis for the categorical outcome of abnormal CBCL, which further adds to the response to the reviewer's comment regarding clinical interpretability of these findings.

Similarly, a new **Figure S4** replicates the analysis for the frequency of snoring as a predictor for adverse categorical CBCL outcomes, therefore supplementing **Figure S3**.

1. Please provide context around CBCL normative data vs these results. The mean scores appear to be within normal range for every 'adverse behaviour'.

As stated in responses to the earlier comments, we provide additional data on the proportions of children with abnormal CBCL scores appended to **Table S1**. In addition, we added the following text to **page 5, para 1, line 100** onwards, “...The proportion of children with abnormal CBCL scores, defined using established thresholds,²² are shown in Table S1. For the internalizing problems scale, 1694 children, or 16.7%, were classified as abnormal based on the threshold t- score of 60 or higher...”

Figure S2 now contains a plot showing the relative risk of abnormal CBCL score showing dose-dependence with the frequency of snoring categorized as non-habitual and habitual. We also replicated the mediation models that incorporated two regressions, the first between oSDB factor score and ROI volumes, and the second modeling CBCL categorical outcome of ‘abnormal score’. Here we show mediation effects (**Figure S2**) similar to **Figure 4** and extend the results to snoring as a predictor in **Figure S3**.

We describe on **page 8, para 3, line 181 onwards**), “...We additionally determined whether the mediation effects shown in **Fig. 4** were also identified in the prediction of children with abnormal CBCL measures meeting the criteria for further diagnostic evaluation. In **Fig. S3a**, we show that the relative risk of abnormal CBCL scores was greater with habitual snoring compared to control children. This elevated risk was also identified for non-habitually snoring children albeit smaller in magnitude. The mediated effects for abnormal CBCL scores related to regional cortical volumes for the top ten cortical regions (**Fig. S3b,c**) were greater in magnitude but regionally conserved compared to **Fig. 4**. For example, smaller volume of right precentral gyrus mediated the association between oSDB factor score and abnormal rule-breaking behavior (% mediated, 4.78 [1.55 to 10.48], $P < 0.001$). The frequency of snoring replicated this specific regional mediation effect (**Fig. S4**, % mediated, 4.58 [1.67 to 11.44], $P < 0.001$)...”

2. How many of this population had abnormal CBCL scores and how do they compare to no SDB populations?

Please note our response to the comment above as the data are presented in Table S1 (shown below).

CBCL category	Mean t-score (\pm SD)	N of children with borderline t-score (%)	N of children with clinical t-score (%)
Anxious/depressed	53.5 \pm 6.0	482 (4.8)	285 (2.8)
Somatic problems	54.9 \pm 6.0	536 (5.3)	286 (2.8)
Social problems	52.7 \pm 4.7	NA	NA
Thought problems	53.8 \pm 5.9	NA	NA
Attention problems	53.8 \pm 6.1	NA	NA
Rule-break behavior	52.5 \pm 4.8	199 (2.0)	217 (2.1)
Aggressive behavior	52.8 \pm 5.4	346 (3.4)	219 (2.2)
Internalizing problems	48.5 \pm 10.6	708 (7.0)	986 (9.7)
Externalizing problems	45.6 \pm 10.2	432 (4.3)	582 (5.7)
Total problems	45.8 \pm 11.2	NA	NA

We also compare the relative risk of children with abnormal CBCL scores between children with history of non-habitual and habitual snoring, showing an increased risk for adverse behavior with the latter (see below for **Figure S4 (a)**)

3. Please direct the reader whether a CBCL score increase of 3.3 units clinically significant?

We thank the reviewer for pointing this out. We have deleted this sentence due to two reasons: First, we removed the reference to the presence of any symptom of oSDB, which may mislead the reader to assume a disproportionately high prevalence of oSDB, as the reviewer stated earlier. Instead, we reported the prevalence of habitual snoring, which is directly comparable to the prevalence of oSDB from pooled studies of habitual snoring. Next, given that we reported habitual snoring as a predictor, we examined (i) its associations with cortical thinning with results in **Figure 3f-i**, (ii) its role in the mediation pathway as shown in **Figure 4d**, and (iii) its prediction of abnormal CBCL scores as shown in **Figure 3a**. We believe these results are more comparable to other studies than presenting the coefficient of the oSDB vs. CBCL regression.

4. Analyses should be repeated using habitual snorers vs sometimes vs occasional (or present snoring on a scale 1-5 as per your Likert scale for data collection) to see if data analyses are replicated when compared to using the composite score. Specifically, it would be helpful to see if ‘snoring more’ has a dose dependent relationship with CBCL and cortical changes.

In all our analyses, we have used snoring first as a continuous predictor. We believe the reviewer may have misinterpreted our mentions of ‘snoring’ as being a categorical predictor due to lack of sufficient clarity. The only instances where snoring has been converted to a categorical predictor are where the reviewer requested contextual comparisons with clinically relevant categories of snoring (**Figures 3, 4**)

We have therefore clarified this in multiple locations:

Figure 3, “...**(a-c)** Atlas-based effect size maps projected on a cortical surface demonstrate the strength of the relationship between the **frequencies of individual symptoms**, the **total oSDB factor score** and **cortical morphometric variables** (average cortical thickness (left) and total volume (right)) measured using structural magnetic resonance imaging (MRI)...”

Figure 4, “...**(c)** shows the top ten mediated effects for cortical regions of interest (ROI) for both oSDB factor score and the frequency of snoring showing similar effects. Most of the mediated effects, especially within the precentral gyri, were observed for ROI volumes and not for thickness or area (not shown)...”

Figure S2, “...**Mediation effects for the relationship between the frequency of snoring and Child Behavior Checklist (CBCL) measures** attributed to regional cortical volumes. Covariate-adjusted effect size maps show the regional mediated effects that define the association between the **frequency of snoring reported on a five-point scale as 1=never, 2=occasional (once or twice a month), 3=sometimes (once or twice a week), 4=habitual (more than twice a week but not daily), and 5=daily. ...**”

Figure S4, “...**Mediation effects for the relationship between the frequency of snoring and abnormal Child Behavior Checklist (CBCL) measures**. Covariate-adjusted effect size maps show the regional mediated effects that

define the association between the frequency of snoring reported on a five-point scale as 1=never, 2=occasional (once or twice a month), 3=sometimes (once or twice a week), 4=habitual (more than twice a week but not daily), and 5=daily...

5. I did not appreciate Fig 4d demonstrated the effect size for frequency of snoring and average cortical thickness rather it was the presence of snoring per se- is that correct?

No, we have presented the snoring data throughout the manuscript as a continuous variable (i.e. the Likert scale recorded in the SDSC subscale used to collect data). Therefore, it the frequency of snoring and not its presence alone (please see above response).

We have clarified this in the figure legend for the current **Figure 3**, "...**(a-c)** Atlas-based effect size maps projected on a cortical surface demonstrate the strength of the relationship between the frequencies of individual symptoms, the total oSDB factor score and cortical morphometric variables (average cortical thickness (left) and total volume (right)) measured using structural magnetic resonance imaging (MRI)..."

6. Would suggest also doing mediation models using habitual snoring only rather than oSDB factor score. That is, evaluate the strength of the relationship between snoring as an independent variable and Cortical morphometry and CBCL as dependent variable (as per Fig 5 data).

We have performed this additional analysis and added it to **Figure 4**, wherein the frequency of snoring largely replicated the mediated effects associated with oSDB factor score (c). We have also added the mediation effects of recategorized

frequency of snoring (panel d) to show a potential threshold effect for habitual snorers.

The effect size brain maps related to the effect of frequency of snoring that follow the results in **Figure 4** are now presented in **Figure S4**.

7. The magnitude (not statistical magnitude) of changes in the cortex, either thickness or volume should be outlined more clearly. Specifically, are these changes clinically relevant?

Our interpretation, in the absence of literature concerning the threshold for loss of brain matter in preadolescents, is largely based on the identifiable loss of cortical gray matter among children with habitual snoring compared to non-snoring children. We deem that there is no safe threshold for cortical thinning, specifically as the relevance of the thinning is largely dependent on the topological function of the cortical regions. Therefore, we provide clinical context for this in the form of panels f-i in Figure 3 and panel d in Figure 4, both of which highlight the fact that children with habitual snoring are at risk and merit further evaluation. Together we believe we provide sufficient clinical context for the translation of these results.

Discussion

1. The discussion highlights well the strengths particularly the large longitudinal dataset especially the collection of MRI data in > 10,000 children.

We thank the reviewer for the comment.

2. While these data are interesting, and considered in the context of other studies, the discussion does not really highlight novelty of these data compared with other studies.

In addition to clarifications in the Introduction, we provide additional emphasis on novelty of the findings in the Discussion (page 11, para 1, line 231 onwards), "...Therefore, these results, derived from simultaneously modeled effects of oSDB on structural alterations in the brain as well as behavioral scores, provide a novel neuroanatomic basis for commonly reported behavioral problems in children with symptoms of oSDB. Furthermore, the observed mediation effects for the relationship between the frequency of snoring and behavioral measures were more readily apparent in children with habitual snoring, supporting the threshold for screening as promulgated by the American Academy of Pediatrics³. However, some loss of statistical power may occur as a result of recategorization of Likert-like scales and hence these results should also be viewed with caution. The evidence for a threshold effect for neuroanatomic changes directly or indirectly attributable to the frequency of snoring may facilitate the refinement of screening practices to identify children with oSDB and are at risk for neurobehavioral morbidity..."

3. Asthma and/or allergy data – if not available, then the lack of this data should be added to limitations.

While asthma was added as a covariate in the appropriate analyses, we chose to not add the allergies, since this was a non-specific characteristic in the ABCD dataset.

Page 13, para 1, line 291 "...Although asthma was included as a covariate in our analyses of the ABCD dataset, there may be other potential unmeasured confounders such as atopic pathology and parental snoring ..."

4. Line 267 onwards in discussion...’the current supports novel common ground to support screening for the condition based on its observations with regional brain structure’.... please clarify this statement

In the United States, the American Academy of Sleep Medicine supports universal polysomnography prior to treatment of oSDB. However, otolaryngologists who perform approximately 500,000 adenotonsillectomies rarely obtain sleep studies prior to treatment of oSDB, reserving it only for children with high-risk conditions such as Down syndrome according to their clinical guidelines. Our results aim to provide support for universal screening for oSDB, regardless of the strategy used, i.e. clinical assessments with or without polysomnography.

Small edits were added for clarity (**page 14, para 2, line 326 onwards**), “...Given that these differences in management strategies are shaped by the interactions between each of these specialties and children with oSDB, the current manuscript provides common ground to **support the screening for and further evaluation of oSDB** based on its observed associations with **regional brain structural alterations...**”

Reviewer #3 (Remarks to the Author):

By analyzing data from a very large cohort of preadolescents (n=10,140), the authors reported strong associations between obstructive sleep disordered breathing (oSDB) symptoms and regional cortical thickness in the brain. Since oSDB is commonly related to neurobehavioral deficits, the findings in this study can help connect brain structural changes to both oSDB and the associated neurobehavioral abnormalities. This novel study is expected to have great interest to the scientific community. The reported results are well supported by sound statistical analyses and the methods are generally rigorous. The study leveraged a large, multi-center imaging database from the ABCD project, comprising not only structural MRI but also functional and diffusion MRI from which brain connectivity information can be derived. It is somewhat puzzling why none of the brain connectivity results, which could be relevant to brain alterations associated with oSDB, was mentioned in the manuscript. Inclusion of brain connectivity information can potentially enhance the conclusions, provided that it is supportive. Another weakness is that the casual aspects of the relationships described in the manuscript are uncertain. A longitudinal study design, as opposed to a cross-sectional design employed in the study, may help provide clarification. Overall, this is a very interesting study with potentially high impact. However, the limitations outlined above, together with additional limitations that the authors discussed in the manuscript, somewhat weakened the conclusions.

We thank the reviewer for the positive comments and agree that these results, along with those we added as part of the revision, are likely to generate serious interest within the scientific and clinical communities. We acknowledge that the current study is limited in that we included morphometric results without connectivity information. The ABCD dataset does contain raw and minimally processed task-activated and resting-state functional MRI that may be used to analyze brain connectivity. However, we believe that the current manuscript provides a more parsimonious conceptual advance in linking the behavioral morbidity associated with oSDB to brain structural changes. Our results show spatial specificity for alterations in the brain regions that shape behavior and cognitive control. Furthermore, we show that these changes are more pronounced with habitual snoring and may be linked additionally to clinical behavioral problems. These data provide a balance of interpretability and novelty within the context of gaps in knowledge.

Analyzing brain connectivity afresh from the minimally processed fMRI datasets from >10,000 children will require major effort and personnel resources, which we hope to accomplish from future grant support. Such a herculean task is incompatible with the timeline of this manuscript. However, it certainly is an area that we would like to focus on in the near future and have therefore stated so in the manuscript (**page 15, para 2**), “...These results are also likely to spur longitudinal models, including that of **other brain measures such as functional connectivity, tissue integrity and task-related functional imaging** within the ABCD cohort and others to investigate oSDB as a potential reversible cause of disruptive **behavior** and poor classroom performance...”

We have already mentioned the limitations owing to the cross-sectional data used in the manuscript (**Page 12**), “...The principal drawback of the current study is its cross-sectional nature. Therefore, none of the analyses described in the manuscript assume directionality or causal aspects of these relationships due to the potential for bias in cross-sectional mediational models. Although our study aimed to partially mitigate this via statistical techniques such as the use of false discovery control in the regional brain changes and bias-corrected bootstrap in the mediation models that reduce model uncertainty, this limitation cannot be fully addressed without longitudinal data³¹...”

Reviewer #1 (Remarks to the Author):

The authors have satisfactorily addressed my comments on their manuscript.

Reviewer #2 (Remarks to the Author):

Thank you for the revised manuscript. The authors have elegantly and comprehensively addressed all my comments. I very much appreciate the time and effort to complete all the revisions.

Reviewer #3 (Remarks to the Author):

In the revised manuscript, the authors have made considerable efforts to address the weaknesses identified by the reviewers. As a result, the completeness and clarity of the manuscript have been noticeably improved. I found the authors' responses are satisfactory and do not have further questions.